# Position: Privacy Is a Claim, Not a Property of Synthetic Data

Jiachen Zhao [1]   Antonia Januszewicz [1]   Taeho Jung [1]

## Abstract

Synthetic data has become a common component of machine learning research. While widely adopted, its use in privacy-sensitive contexts has quietly shifted from a claim of residual inference risk under stated assumptions to an appearance-based property inferred from data generation itself. In this position paper, we argue that this shift reflects an implicit change in community standards for what counts as sufficient privacy evidence, rather than a misunderstanding of well-established privacy principles. Drawing on an empirical analysis of recent publications across major ML venues, we show that synthetic data is frequently used in privacy-sensitive settings without explicit articulation of threat models, inference risks, or falsifiable privacy claims. As a result, privacy assurance often remains implicit, difficult to verify, and unevenly distributed, with heightened exposure for rare and minority records. We argue for treating privacy as an explicit, evidence-based scientific claim and recommend that ML venues adopt norms requiring privacy-relevant assertions to be clearly scoped, testable, and contestable.

## 1. Introduction

Synthetic data is increasingly treated as a default response to privacy concerns in machine learning. Across research, industry, and policy, the generation of non-real data is widely assumed to mitigate or resolve privacy risks, enabling data sharing, benchmarking, and deployment without directly engaging adversarial inference. This assumption is rarely formalized, yet it shapes how privacy language is interpreted and trusted: synthetic data is often described as "privacy-preserving" or "safe by construction" based on how it is produced and presented rather than on verifiable and formal guarantees. As a result, privacy is no longer treated

primarily as an empirical claim about inference risk, but as an inferred property of generation itself. This shift has occurred gradually and largely without debate, becoming embedded in publication norms, vendor claims, and institutional interpretations of privacy.

We argue that this appearance-based framing of privacy is structurally unsound. When privacy assurance is inferred from surface properties—such as non-identity, novelty, or the absence of direct replicas—dominant leakage channels are systematically overlooked. Modern generative models can encode training dependence without reproducing identifiable records, allowing inference risk to persist even when standard audits succeed. Moreover, synthetic data does not eliminate privacy risk but redistributes it: probability mass concentrates around common patterns, while rare or atypical records become disproportionately exposed. These effects are obscured by aggregate evaluation and average-case reasoning, producing false negatives that are not incidental but inherent to how privacy is assessed. As a consequence, privacy claims become difficult to contest, compare, or falsify once generation has occurred, even when meaningful inference risk remains.

This paper argues that privacy in machine learning must be re-grounded as an explicit, verifiable, evidence-based claim rather than an implicit property of synthetic data. Synthetic generation may reduce exposure and enable access, but it cannot, on its own, establish privacy. We argue for recalibrating how privacy claims are articulated and evaluated: when synthetic data is used to motivate or justify privacy, it must be accompanied by mechanisms that yield interpretable risk knowledge. We engage alternative views that treat synthetic data as a pragmatic risk-reduction measure or rely solely on disclosure, and explain why these approaches cannot bear the epistemic weight currently placed on them. Finally, we call for lightweight yet coordinated shifts across research practice, review standards, and institutional interpretation to prevent synthetic data from serving as de facto privacy evidence and to restore verifiability to privacy claims in ML.

Concretely, this paper:

(i) characterizes the conceptual shift from privacy as adversarial risk to privacy as appearance;

(ii) provides an empirical audit of ICML, NeurIPS, and

---

[1]Department of Computer Science Engineering, University of Notre Dame, Notre Dame, United States. Correspondence to: Taeho Jung <tjung@nd.edu>.

*Proceedings of the $43^{rd}$ International Conference on Machine Learning*, Seoul, South Korea. PMLR 306, 2026. Copyright 2026 by the author(s).

ACL (2024–2025) showing that synthetic data is widely used in privacy-sensitive settings while verification remains rare (Section 2.3);

(iii) maps existing tools to the resulting failure modes and distills them into a **Minimum Privacy Claim Standard** (Section 4.2).

# 2. The Conceptual Shift: From Formal Privacy to Empirical Privacy

## 2.1. From Privacy as Risk to Privacy as Appearance

Before using synthetic data became a mainstream practice, privacy in machine learning was articulated using adversarial risk claims. In an adversarial risk claim, one specifies a threat model and then argues (via bounds or attacks) about individual-level inference. Under this view, privacy guarantees cannot be made based on the properties of a released dataset; what matters is residual inference risk given the adversary's auxiliary information and inference methods (Dwork et al., 2014; Clifton & Tassa, 2013; Narayanan & Felten, 2014; Brown et al., 2022). In recent years, particularly with the rise of generative models, privacy has increasingly been addressed in a different way: it is treated as something that follows from how an output *looks* or how it was *generated*. While LLM-generated synthetic text serves many purposes—such as data augmentation, robustness testing, and prototyping—it is increasingly treated as privacy-preserving by default in data-sharing contexts. A common argument is that synthetic data justifies privacy through non-identity: because synthetic records are generated rather than directly copied, the absence of a one-to-one correspondence with real, sensitive data is taken to imply privacy (Bellovin et al., 2019). Under this framing, removing direct identifiers or exact replicas is sufficient evidence that privacy risk is mitigated (Gal & Lynskey, 2023). As such, privacy is treated as a property of the output artifact, assessed in terms of non-replication (no verbatim matches), transformation, or statistical plausibility, rather than through explicit reasoning about what an adversary could infer.

This output-centric interpretation of privacy has become particularly popular in the era of large language models (LLM). LLMs can generate synthetic text that is fluent, diverse, and rarely verbatim, satisfying many of the surface-level criteria commonly associated with unidentifiability. Because such outputs usually lack explicit record-level correspondence to sensitive information through construction or prompts, they are often treated as inherently privacy-preserving. Theoretically, this intuition aligns with views that characterize generative models as sampling from a learned distribution rather than retrieving individual training examples (Brown et al., 2022). Empirically, early evaluations of LLM privacy have emphasized memorization and exact reproduction as primary risk signals, reinforcing the notion that absence of verbatim leakage constitutes meaningful privacy protection (Carlini et al., 2021). Together, these factors make LLM-generated synthetic data falsely appear well-suited to serve as privacy-preserving substitutes, even in the absence of explicit threat models or bounded inference claims.

## 2.2. Industry, Policy, and Legal Narratives Reinforcing the Shift

To understand why appearance-based privacy claims persist in ML practice, it is necessary to examine the incentive structures that shape how privacy is evaluated and justified across the deployment pipeline.

Industry-facing documentation plays a central role in normalizing an output-centric understanding of privacy. Commercial synthetic data vendors routinely frame synthetic outputs as "not personal data" or "anonymous by construction," grounding privacy claims in the absence of direct identifiers or record-level correspondence, branding synthetic data as an efficient and economic alternative to sensitive real-world data (Mostly AI, 2020; MDClone, 2022). Table 1 provides an illustrative rather than exhaustive snapshot of how privacy claims are articulated across representative commercial synthetic data tools. As the table shows, many tools either mention differential privacy at a high level or omit formal privacy mechanisms (Gallio, 2025; NVIDIA, 2024). This framing is economically attractive within ML pipelines because it preserves utility while minimizing compliance overhead. Vendors emphasize non-identity alongside statistical fidelity, treating generation itself as a sufficient transformation and presuming privacy without explicit adversarial analysis.

As reflected in Table 1, verification is frequently limited to inexpensive, easy-to-automate, and easy-to-communicate checks: testing for verbatim reproduction, citing high-level privacy mechanisms, or relying on certifications that assess organizational controls rather than inference risk. While such checks can rule out narrow failure modes, they do not bound membership, attribute, or distributional inference under realistic adversaries (Bellovin et al., 2019; Carlini et al., 2021; Houssiau et al., 2022). More fundamentally, the limited adoption of strong anonymization and risk-bounding techniques reflects an incentive misalignment rather than a technical limitation. Methods that preserve realism and downstream performance while minimizing verification and governance burden are systematically favored over those that provide explicit, adversary-aware guarantees (Schneider et al., 2025). As a result, privacy claims tend to stabilize around procedural sufficiency rather than demonstrable limits on inference. Recent interdisciplinary assessments have explicitly warned that synthetic data is particularly vulnerable to such misuse, as privacy claims can be sustained even

*Table 1.* Privacy Claims in Commercial Synthetic Data Tools (more details in Appendix A)

| Tool | Use Case | DP Usage | Verification | Privacy guarantees |
|------|----------|----------|--------------|-------------------|
| Tonic.ai[1] | Dev/test data | Mentions DP only | None | no guarantees |
| Hazy (SAS)[2] | Healthcare/finance | Mentions DP only | Unverified | no guarantees |
| YData[3] | ML pipelines | DP in sampling | Internal tests | partial DP |
| Gretel[4] | General ML data | Optional DP filters | Internal + GCP | configurable |
| Syntho[5] | Healthcare/finance | No DP, externally reviewed | Third-party eval | audited |
| MOSTLY AI[6] | Enterprise data | Optional DP | SOC2 + ISO 27001 | infrastructure level |
| DataCebo (SDV)[7] | Academic/research | Explicit DP (with $\varepsilon$) | Bundled DP audit | quantifiable DP |

*Note:* SOC 2 and ISO 27001 ensure secure handling of data, but do not verify synthetic data privacy.
SOC 2: Assesses controls for data security and privacy.
ISO 27001: Requires risk-managed information security practices.

when implementation details and verification procedures remain opaque (De Cristofaro et al., 2025).

Legal scholarship documents how this output-based intuition migrates into regulatory interpretation, further stabilizing appearance-based privacy claims. Across multiple analyses, synthetic data is treated as categorically distinct from personal data when it lacks explicit linkability, shifting the legal inquiry away from residual inferability and toward formal classification (Gal & Lynskey, 2023; Cofone et al., 2024). Under this approach, the privacy status of synthetic data is determined primarily by its detachment from identifiable persons, rather than by what attributes, memberships, or population-level inferences remain possible under realistic adversarial conditions. This framing stands in tension with earlier data protection doctrines, which emphasized contextual unidentifiability, auxiliary information, and potential re-identification as the relevant legal tests (Weitzenboeck et al., 2022).

Crucially, the alignment between industry practice and legal classification reflects a structural dynamic that shapes how privacy is evaluated within ML systems rather than a mere conceptual disagreement. European data protection authorities explicitly reject blanket claims of anonymity and emphasize case-by-case assessment of residual inference risk, memorization, and indirect identification (EDPS, 2021; EDPB, 2024), yet these frameworks rarely mandate adversarial audits, explicit threat models, or quantitative risk bounds. Recent U.S. statutory language increasingly classifies synthetic data as non-personal when it lacks direct identifiers, without requiring additional justification or inference-based evaluation (Utah State Legislature, 2024). A recent work argues that techniques framed as advancing "AI safety" or "privacy" can function as mechanisms of regulatory avoidance, signaling compliance while relocating systems outside traditional oversight (Yew & Judge, 2025).

The issue for ML research is not regulatory noncompliance, but the normalization of evaluation regimes that treat privacy as satisfied by generation or transformation alone. This framing influences how privacy claims are articulated, evaluated, and trusted in contemporary ML research.

### 2.3. Synthetic Data as an Implicit Privacy Claim in ML Research

The appearance-based framing of privacy is also evident in contemporary machine learning research practice, where synthetic data often serves as an implicit privacy justification. Across major ML venues, synthetic data is often introduced in contexts where privacy, safety, or responsible data use is cited as a motivating concern, even when no explicit privacy mechanism is specified. In such cases, the use of synthetic data itself appears to stand in for adversarial analysis or formal guarantees.

To characterize this pattern, we conduct a descriptive audit of accepted papers from ICML, NeurIPS, and ACL (2024–2025), applying a conservative, rule-based screening pipeline to published PDFs, designed to undercount rather than overcount appearance-based claims[1]. The pipeline identifies (i) explicit or indirect use of synthetic data, (ii) the presence of privacy-related language, and (iii) references to formal or empirical privacy mechanisms such as differential privacy, adversarial audits, or bounded guarantees. Importantly, the analysis does not evaluate correctness or safety; it documents how privacy is framed and supported in current research practice. For completeness, we defer implementation details and filtering criteria to Appendix B.

Figure 1 summarizes the prevalence of synthetic data usage across venues and years, distinguishing between explicit

---

[1]Code and data-processing scripts are available at https://anonymous.4open.science/r/synthetic-8735/.

synthetic data construction and weaker, indirect forms such as bootstrapped or pseudo-labeled supervision. While absolute rates differ by field, synthetic data usage is consistently present across years and venues, indicating that synthetic data has become a routine component of contemporary ML research workflows rather than an exceptional or specialized technique.

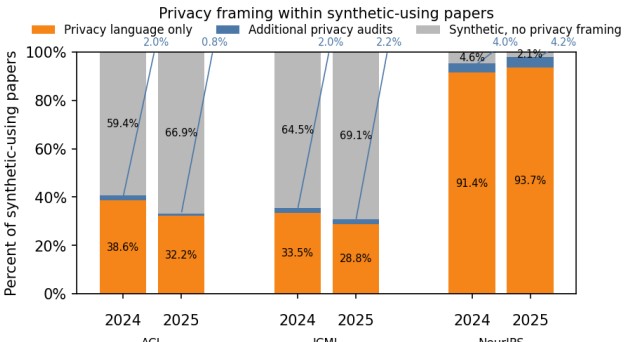

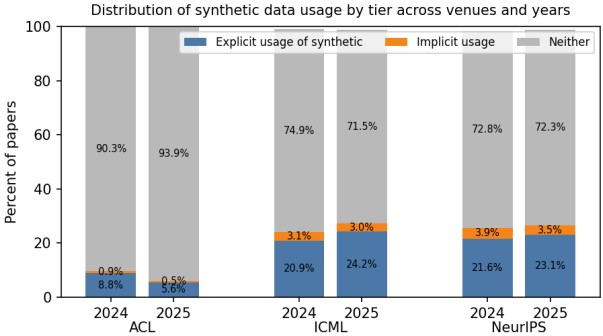

*Figure 1.* Prevalence of synthetic data usage across venues and years, distinguishing explicit synthetic data construction (Tier 3) from indirect or implicit forms(Tier 2). Simulation-only papers are excluded.

Figure 2 conditions on papers that use synthetic data and examines how privacy is articulated in those cases. Across venues, a substantial fraction of synthetic-using papers invoke privacy-related motivations, such as reduced exposure or safer data sharing, without pairing these claims with explicit verification. At NeurIPS, the near-universal presence of privacy language is largely attributable to mandatory ethics and reproducibility statements rather than to privacy being a central claim of the work.

To make this imbalance more concrete, Figure 3 zooms in on ICML 2025 and shows the composition of privacy framing among synthetic-using papers. The result shows that only a small minority of synthetic-using papers pair privacy-related statements with formal guarantees or adversarial evaluation. This pattern indicates that the presence of privacy discourse alone is a weak indicator of substantive risk accounting. Papers that make no privacy claims fall outside the scope of our argument, as they may use synthetic data purely for augmentation, debugging, or benchmarking; though some may be making such claims implicitly. For more information on how the data was gathered, see Appendix B.

Additionally, these figures illustrate a consistent shift in how privacy is handled in ML research. Synthetic data is increasingly used in privacy-sensitive contexts, yet privacy is more often implied by generation than specified through explicit assumptions, threat models, or failure modes. This characterization is descriptive rather than evaluative: it does

*Figure 2.* Privacy framing and verification among papers that use synthetic data. Bars show the fraction of synthetic-using papers that (i) do not invoke privacy, (ii) invoke privacy-related language without accompanying verification, or (iii) pair privacy language with explicit verification mechanisms such as audits or formal guarantees.

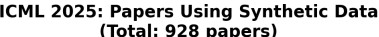

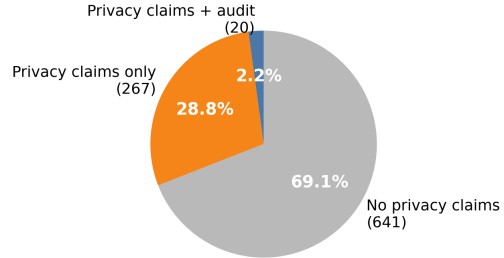

*Figure 3.* Breakdown of Privacy language and verification used in synthetic-using Papers (ICML 2025)

not assert that synthetic data is ineffective or unsafe, nor that risk is not reduced in practice. Rather, it highlights a structural pattern in which generation is treated as sufficient evidence of privacy, even when no falsifiable privacy claim is articulated.

This pattern aligns closely with current conference guidance. Author instructions for ICML, NeurIPS, and ACL emphasize responsible data collection, transparency about data sources, and compliance with applicable laws, but they do not specify how privacy claims should be supported when synthetic data is used in sensitive contexts (International Conference on Machine Learning, 2026; Neural Information Processing Systems, 2025; Association for Computational Linguistics, 2025). Papers that rely on synthetic data in privacy-sensitive settings, therefore, satisfy venue expectations without needing to specify how potential leakage or downstream misuse is assessed. This alignment between venue guidance and research practice helps explain why privacy is frequently justified solely on the basis of data generation, without any explicit deviation from stated conference standards.

# 3. Consequences of Not Correcting the Shift

The reframing of privacy as a property of synthetic outputs rather than as an adversarial risk does not merely weaken individual evaluations; it induces a systemic failure mode. Once privacy is inferred from generation itself, downstream practices—auditing, harm assessment, governance, and accountability—reorganize around that assumption. The result is a privacy regime that appears functional while silently losing its ability to detect, measure, and respond to harm. This section traces three interlocking consequences of that shift.

## 3.1. False Negatives Become the Norm

When privacy is evaluated through the inspection of released synthetic data, audits tend to focus on what is most directly observable. Common practices test whether synthetic samples resemble specific training records, whether injected secrets reappear, or whether similarity-based metrics exceed predefined thresholds. These evaluations are attractive because they are intuitive and easy to implement. However, they implicitly assume that privacy risk manifests through visible reproduction.

Recent work shows that this assumption is systematically misaligned with how leakage occurs in modern generative models. Synthetic data can pass reproduction- and similarity-based audits while still encoding exploitable dependence on the training distribution. Prior work shows that membership signals can persist through reconstruction error, density structure, and local overfitting even when synthetic samples appear non-identifying (Meeus et al., 2025; van Breugel et al., 2023; Ward et al., 2025). As a result, the absence of detectable secrets often reflects output filtering rather than the absence of training dependence.

Empirical studies demonstrate how this mechanism manifests across architectures. In diffusion models, membership can be inferred from systematically lower reconstruction error on training records even when no verbatim copies appear (German et al., 2025; Cheng & Bahmani, 2025). Statistical generators exhibit analogous failures: preserving marginal or structural correlations exposes rare or high-information individuals, and tailored attacks remain effective even with limited attacker knowledge (Golob et al., 2025; Andrey et al., 2025). Similar distributional leakage appears in large language models, where in-distribution canaries leak at far higher rates than random strings (Meeus et al., 2025). Downstream use further amplifies these effects—fine-tuning on synthetic data increases both PII extraction and membership inference accuracy—demonstrating that synthetic data can propagate, rather than attenuate, training dependence (Akkus et al., 2025).

Crucially, these failures arise in regimes where surface-level audits appear clean. Similarity- and distance-based evaluations can remain well within acceptable ranges even when membership inference succeeds at scale, indicating that positive audit outcomes do not bound inference risk (Yao et al., 2025; Pilgram et al., 2025). This is because scalable attacks exploit distributional training dependence rather than memorizing identifiable records, recovering a membership signal without verbatim overlap or explicit reproduction (Nasr et al., 2025). As a result, audits often certify surface properties of the output distribution—plausibility, novelty, smoothness—while statistically detectable traces of training data remain accessible to adversaries (Shi et al., 2024). The resulting false negatives are therefore structural, not incidental.

Taken together, these findings indicate that false negatives are not accidental outcomes of poor auditing but rather a predictable consequence of evaluating privacy based on output appearance. When claims of privacy are based on the appearance of synthetic data rather than on what can be inferred from it, undetected leakage becomes the norm rather than the exception. (Shanmugarasa et al., 2025).

## 3.2. Risk is Redistributed, Not Eliminated

A second structural consequence of appearance-based privacy is that synthetic data frequently redistributes privacy risk rather than eliminating it. Generative models are trained to maximize likelihood under the empirical data distribution, which favors accurate modeling of high-density regions while providing limited support for low-density regions. As a result, common patterns are absorbed into the background of the synthetic distribution, whereas rare or distinctive records become statistically significant. Privacy protection under this regime is therefore inherently asymmetric.

This asymmetry has been repeatedly documented in inference attacks against generative models. Records located in low-density regions of the data distribution, corresponding to rare attribute combinations or atypical cases, carry higher information content and are therefore more likely to be memorized or reconstructed during training (Hayes et al., 2019; Hilprecht et al., 2019). Subsequent analyses formalize this effect by showing that density-based inference disproportionately succeeds on such tail records, even in the absence of verbatim reproduction (Zhang et al., 2022; Ward et al., 2025).

Since standard audits aggregate risk across the dataset and emphasize average-case behavior, this concentration of harm remains largely unobserved. Analyses of synthetic data evaluation practices show that aggregate utility, similarity, and diversity metrics systematically mask subgroup-level exposure, creating false confidence even when tail records experience elevated risk (Ravn, 2024; Yao et al., 2025; Pilgram et al., 2025). Privacy exposure is not elimi-

nated but shifted toward specific regions of the distribution.

This redistribution has systematic fairness implications. Individuals associated with rare attribute combinations, often corresponding to minority groups, edge populations, or sensitive sub-cohorts, are simultaneously the hardest to model accurately and the easiest to re-identify statistically. Empirical studies of synthetic tabular and health data show that membership inference success rates are substantially higher for low-frequency records, even when global utility and similarity metrics appear favorable (Zhang et al., 2022; van Breugel et al., 2023; Johnson & Hajisharif, 2024). Work on diversity-washing and synthetic augmentation further demonstrates that synthetic data can give the appearance of protection or representational improvement while reproducing or amplifying harms for already underrepresented groups (Whitney & Norman, 2024). Because these effects are diluted in aggregate evaluations, they are unlikely to be detected or reported under appearance-based auditing.

Increasing generation fidelity does not resolve this issue and can, in fact, intensify it. Higher-fidelity synthesis preserves fine-grained correlations more accurately, strengthening the statistical signals available for inference. In language and multimodal models, this appears as distributional or semantic leakage, where synthetic outputs have surface-level non-identifiability yet retain stable associations that enable attribute inference or population-level disclosure (Staab et al., 2024; Li et al., 2024). Such leakage reflects the consequences of optimizing for realism under constrained data regimes rather than deficiencies in generation quality.

Overall, synthetic data does not provide uniform privacy protection. Instead, it reallocates risk across the distribution, reducing exposure for typical cases while increasing exposure in the tails, without making this redistribution explicit. Appearance-based privacy, therefore, fails structurally, resulting in uneven and largely undetected privacy loss.

### 3.3. Institutional Consequence: Privacy Becomes Non-Actionable

The primary consequence of appearance-based privacy is institutional. When privacy claims are grounded in the use of synthetic generation itself, rather than in explicit guarantees or threat models, they become non-falsifiable. There is no clear definition of failure beyond the absence of direct identifiers, no shared metric for residual risk, and no standard by which claims can be challenged. Privacy becomes a descriptive label rather than an enforceable property.

The shift weakens accountability mechanisms that rely on comparability and contestability. Privacy assurances that lack explicit assumptions about adversaries or inference pathways cannot be meaningfully compared across datasets or evaluated over time. Empirical studies of data and model

documentation show that downstream users often lack visibility into the provenance of training data, retention behavior, or evaluation scope, further limiting the ability to interrogate privacy claims (Roberts et al., 2024; Jacovi et al., 2023).

In the absence of mandated standards, synthetic-data audits often certify privacy through narrow or self-selected criteria, emphasizing procedural compliance rather than residual risk. Privacy auditing research shows that, without standardized threat models or success conditions, auditors retain wide discretion over what counts as evidence, enabling positive conclusions that are technically correct yet substantively uninformative (Stadler et al., 2022). Empirical studies confirm that synthetic data can satisfy commonly reported audit checks while remaining vulnerable to singling-out or membership inference, not because risk is absent, but because it falls outside the audit's chosen scope (Meeus et al., 2025). In this regime, audits do not fail silently; they succeed in ways that leave privacy claims effectively non-falsifiable.

Legislative and regulatory asymmetries further entrench this dynamic. Legal scholarship documents a growing tendency to treat synthetic data as categorically non-personal, exempting it from privacy obligations without requiring substantive assessment of residual inference or group-level risk (Veale & Borgesius, 2021; Gal & Lynskey, 2023). Legislative carve-outs that classify synthetic data as outside the scope of personal data remove data-subject rights to access, correction, or deletion, leaving affected individuals without recourse even when downstream harms arise (Utah State Legislature, 2024). Law and policy analyses note that synthetic data can enable inferential and group-level harms, including inaccurate or discriminatory downstream inferences, even when individual linkage is unlikely (Organisation for Economic Co-operation and Development, 2025). In these regimes, privacy is denied not through demonstrated safety, but through definitional exclusion.

Regulatory guidance explicitly rejects this presumption. Data protection authorities in the EU and UK emphasize that synthetic data and generative model outputs cannot be assumed anonymous by default and must be evaluated on a case-by-case basis with respect to residual inference and group-level risk instead (EDPB, 2024; EDPS, 2021). Standards and risk-management frameworks similarly stress that data transformation alone does not eliminate privacy risk and that ongoing assessment is required when outputs retain statistical dependence on individuals or groups (Decentriq, 2025; NIST, 2023). The divergence between statutory exemptions and regulatory guidance creates a governance gap in which privacy claims are asserted, audited, and deployed without a clear locus of responsibility for validating their substance.

In short, these dynamics render privacy non-actionable at the

institutional level. Audits lack standardized success criteria, regulatory oversight is fragmented, and affected parties lack standing because harm is denied at the definitional stage. Restoring accountability requires standardized, independent audit frameworks and mandatory reporting mechanisms that make privacy claims falsifiable by linking disclosure to explicit risk assumptions and verifiable evidence. Without such mechanisms, appearance-based privacy will continue to support privacy claims that cannot be meaningfully audited, compared, or enforced.

## 4. Re-grounding Privacy Claims in Synthetic Data

### 4.1. Tools That Address the Failures

Section 3 showed that synthetic data does not eliminate privacy risk, but redistributes it in ways that are often invisible under current evaluation practices. Importantly, this does not imply that synthetic data is misguided or unusable. It implies that synthetic generation addresses only a subset of privacy concerns, primarily those related to direct data access, while failing to address the fundamental risk of inference, for which established privacy-preserving frameworks already provide solutions.

- **Explicit risk bounding.**

  *Differential privacy* offers a clear example of how privacy can be expressed as a scoped empirical claim rather than a property of data transformation. By defining privacy loss as a parameterized bound on adversarial inference, DP clarifies what protection is provided and under what assumptions (Dwork et al., 2014; Clifton & Tassa, 2013; Abadi et al., 2016; Mironov, 2017). While DP is not universally applicable, its framing illustrates how privacy claims can be articulated in terms of residual risk rather than appearance. This distinction is central to critiques of synthetic data that emphasize the absence of quantified limits in generation-only approaches (Bellovin et al., 2019; Gal & Lynskey, 2023). Differential Privacy can be used to address false negatives in 3.1, non-falsifiability in 3.3. It can also address the uneven risk allocation in 3.2, though only if carefully applied: pure DP bounds worst-case privacy loss for all records, including the low-density tail, whereas approximate $(\varepsilon, \delta)$-DP permits a $\delta$ probability of failure that can concentrate on precisely the rare and minority records most exposed under appearance-based auditing.

- **Empirical inference evaluation.**

  A second class of tools focuses on directly testing whether sensitive inference remains possible from synthetic outputs. Membership and attribute inference attacks against generative models demonstrate that nonidentifiable-looking data can still encode sensitive signals (Hayes et al., 2019; Chen et al., 2020). Audit-oriented frameworks such as TAPAS provide systematic evaluation, allowing privacy claims to be empirically challenged rather than assumed (Houssiau et al., 2022). More recent benchmarking efforts emphasize measuring privacy leakage as a continuous quantity rather than a binary label, enabling comparison across generators and settings (Sidorenko et al., 2025). This class can be used to address non-falsifiability in 3.3, and, if carefully applied, also address the false negatives in 3.1.

- **Risk heterogeneity analysis.**

  A third set of tools addresses the uneven distribution of privacy risk that synthetic data often induces. Empirical studies in health and tabular data show that inference risk concentrates on rare or atypical records, even when average-case metrics appear benign (Zhang et al., 2022). Correlation inference attacks further demonstrate that group-level attributes may remain inferable despite the appearance of individual-level protection (Crețu et al., 2024). Disaggregated reporting aligns with established ML practice, where aggregate performance is no longer treated as sufficient evidence (Arora et al., 2025). These tools can be used primarily to address the uneven risk allocation discussed in 3.2. It could catch the false negatives in 3.1 and challenge the non-falsifiability discussed in 3.3 on a case-by-case scenario.

These tools do not replace synthetic data. They address the specific failure modes identified in Section 3, with different focuses. Their relevance lies in making privacy claims interpretable rather than in enforcing any particular protection strategy.

### 4.2. Privacy Claims are Earned, not Implied

Synthetic data may be used for many purposes—such as exploratory analysis, debugging, or access control—without invoking privacy claims at all. However, when synthetic data is presented as privacy-preserving, the claim should be treated as an empirical assertion rather than an inherent property of the generation process. In this setting, privacy claims should be earned: supported by falsifiable evidence, clearly scoped assumptions, and documented limitations.

This distinction mirrors how other claims are handled in ML research. Robustness is not inferred from architecture choice, fairness is not inferred from dataset curation, and safety is not inferred from intent. In each case, methods may be perfectly valid without additional evaluation, but claims only acquire meaning when paired with appropriate

evidence and scope conditions. Methods that do not engage such evaluation are not prohibited, but they do not enjoy the same interpretive status as those that do.

The tools outlined in Section 4.1 provide ways to articulate what protection is claimed, under what assumptions, and with what limitations. Protocols that apply such tools earn the validity to make interpretable privacy claims. Protocols that do not apply such tools remain scientifically valid, but should not inherit the same implicit privacy guarantees simply by virtue of using synthetic data. Framed this way, the issue is not the use of synthetic data, but the conditions under which privacy claims are treated as meaningful within ML research.

To make this concrete, we summarize the principle as a minimal disclosure standard: authors should state whether synthetic data is used as (a) a risk-reduction heuristic, or (b) evidence supporting a privacy claim. If (a), no further privacy justification is required. If (b), the claim should satisfy three conditions:

1. **Threat model.** State the concrete threat(s) the claim protects against—e.g., the adversary assumed and the inference being bounded.

2. **Scope.** Clarify which risks are addressed and which remain unaddressed.

3. **Evidence.** Provide at least one evidentiary mechanism— e.g., a DP guarantee with stated $\varepsilon$, an inference-based audit, or disaggregated risk reporting.

We refer to these conditions as the Minimum Privacy Claim Standard. The standard introduces no new infrastructure; it asks only that existing tools be applied and stated explicitly, mirroring shifts already adopted elsewhere in ML research.

A healthier norm should not restrict the use of synthetic data but rather clarify its role. Synthetic data may reduce exposure and enable access, but privacy claims must be stated as claims about residual risk, with scope and limitations made explicit. This completion of existing ML norms preserves methodological flexibility while preventing privacy by default through generation.

## 5. Alternative Views

Several alternative perspectives challenge the framing and implications of our position. We outline the most salient ones below, not to dismiss them, but to clarify the scope and intent of our argument.

1. **Synthetic data as pragmatic risk reduction.** One view holds that synthetic data is best understood as a practical mechanism for reducing direct exposure to sensitive records, rather than as a source of formal privacy guarantees. In many applied settings, its value lies in enabling development and limited sharing under access constraints, and requiring explicit audits or risk quantification may be impractical.

   We agree that synthetic data often serves this operational role. Our concern is not with its use as a risk-mitigating heuristic, but with how this heuristic is increasingly interpreted as evidentiary when privacy is invoked as a scientific motivation.

2. **Negative evidence as sufficient in practice.** A second view argues that, in the absence of detected memorization or re-identification, synthetic data can reasonably be treated as lower risk than raw data. As with other evaluation practices in ML, privacy assessments are necessarily incomplete, and negative empirical results may be the most tractable signal available.

   Although such evidence is informative, we argue that its institutional role has shifted from a provisional signal to an implicit closure. When the absence of observed leakage is treated as a resolution of privacy concerns, residual uncertainty is no longer made explicit.

3. **Disclosure norms instead of verification (partially agreed).**

   A third perspective emphasizes clearer disclosure over formal verification, suggesting that explicitly stating what synthetic data does and does not protect against may be sufficient to prevent over-claiming.

   We agree that improved disclosure is necessary and would meaningfully reduce ambiguity. However, our position is that disclosure alone cannot specify which risks are reduced and how, and therefore would be incomplete. Synthetic data may reduce privacy risk by preventing direct one-to-one correspondence or modifying distributional properties, offering qualitatively different risk reductions. As a result, disclosure norms cannot distinguish among types and degrees of risk reduction and therefore cannot prevent synthetic data from becoming de facto privacy evidence institutionally.

Taken together, these views underscore that synthetic data occupies meaningful roles across research and practice. Our position draws a boundary: when synthetic data is used to motivate or justify privacy claims in scientific work, the absence of explicit risk accounting creates systematic blind spots. Clarifying this boundary is necessary to preserve both the usefulness of synthetic data and the meaning of privacy claims in ML research.

# 6. Call to Action

We argue for coordinated but lightweight shifts across the ML research ecosystem to prevent synthetic data from functioning as implicit privacy evidence. *At the research level*, authors should explicitly state whether synthetic data is used as a risk-reduction heuristic or as evidence supporting privacy claims, and whether privacy risk is being inferred rather than evaluated, following the **Minimum Privacy Claim Standard** stated in Section 4.2. *At the conference level*, reviewers and area chairs should treat synthetic data generation as insufficient to establish privacy on its own, and should not interpret its use as resolving privacy concerns absent mechanisms that produce explicit risk knowledge. *At the institutional and deployment level*, organizations and regulators should resist categorical interpretations of synthetic data as "privacy-safe," and require that privacy claims grounded in synthetic data specify what risks remain unaccounted for. These steps formulate a simple norm: **generation may reduce exposure, but it cannot substitute for verification.**

As synthetic data is increasingly deployed in regulated settings for many different use cases, there is growing uncertainty about when its use constitutes a privacy claim. In the absence of shared definitions, privacy claims for synthetic data are interpreted inconsistently across research, deployment, and governance contexts. As the necessary level of anonymization or privacy protection may differ across data types, that information needs to be explicitly articulated. We therefore call for clearer, uniform definitions of what constitutes a privacy claim in the context of synthetic data, as a prerequisite for applying the expectations outlined above.

If one rejects the need for explicit verification, an alternative is to treat synthetic data purely as an access-enabling or exposure-reducing practice, while refraining from invoking privacy-related scientific or institutional claims.

# Acknowledgments

This research was supported by the National Science Foundation under CNS-2337321, OAC-2312973. The authors used Gemini, Grammarly, and ChatGPT for minimal revision of the text to correct typos, order of wording, and grammatical errors. Cursor is used when coding the data-processing codebase. Additionally, we thank Professor Meng Jiang at the University of Notre Dame for his valuable input and guidance on the direction of this position paper.

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

## A. Verification and Privacy Guarantees of Synthetic Data Tools

This appendix provides the evidence supporting the privacy verification categories in Table 1. It summarizes how commercial synthetic data tools define privacy, whether those claims are verifiable, and what level of privacy guarantees they offer.

### A.1. Evidence from Vendor Websites

*Disclaimer: The following summaries are based on vendor documentation and public webpages available at the time of writing. Product offerings and privacy guarantees may change with future updates.*

- **Tonic.ai**
  "Enabling differential privacy will add noise and may remove rare categories."
  "Differential privacy is disabled by default."
  DP toggle configurable per column; $\varepsilon = 1$ by default in some generators.
  *No external audit available.*

- **Hazy (via SAS)**
  Privacy risk measured using density disclosure and presence disclosure metrics.
  "High score close to 1 indicates low risk" in membership-inference simulations.
  *No external audit; internal-only metrics for risk estimation.*

- **YData**
  Claims synthetic data provides "privacy compliance" and "protection against re-identification."
  Uses "automated quality and privacy control," including divergence metrics and inference attacks.
  Offers $\varepsilon$-configurable differential privacy modes (high fidelity, balanced, high privacy).
  *No external audit or certification.*

- **Gretel**
  Features "built-in privacy filters and models with differential privacy."
  Provides "tunable privacy with mathematical guarantees."
  Includes privacy scoring dashboard, integrated with Google Cloud.
  *No third-party audit documented.*

- **Syntho**
  Synthetic data "approved by SAS data experts" for downstream model quality and safety.
  Emphasizes data utility: models trained on synthetic data match those trained on real data.

*No formal mention of differential privacy or privacy audit.*

- **MOSTLY AI**
  Ensures "no one-to-one relation between original and synthetic data."
  All privacy settings "turned on by default."
  Certified under SOC 2 and ISO 27001.
  *No differential privacy, but strong infrastructure and operational controls.*

- **DataCebo (SDV)**
  "$\varepsilon$-Differential privacy" supported with tunable $\varepsilon$.
  "Empirically measures differential privacy" via bundled evaluation tools.
  *Formal privacy guarantee with audit support through quantifiable metrics.*

## Interpretation of Verification and Guarantee Categories

**None:** Vendor makes no mention of tests or audits; privacy claims rely solely on the generative process. Example: *Tonic.ai (without DP)*.

**Unverified:** Vendor claims to use privacy-enhancing technology (e.g., DP or risk metrics) but provides no independent audit or evidence. Example: *Hazy/SAS*.

**Internal tests:** Vendor runs proprietary privacy and utility checks (e.g., divergence metrics, inference attacks), but lacks external validation. Example: *YData*.

**Internal + partner:** Combines built-in privacy filters or scores with deployment on trusted platforms. Example: *Gretel on Google Cloud*.

**Third-party evaluation:** Evaluation of data quality and/or privacy conducted by an external entity (not a formal audit). Example: *Syntho evaluated by SAS*.

**Security certifications:** Vendor holds organizational-level security credentials (e.g., SOC 2, ISO 27001), but no guarantees of data-level privacy. Example: *MOSTLY AI*.

**Bundled DP audit:** Offers a formal $\varepsilon$-DP mechanism and tools to empirically verify privacy loss. Example: *DataCebo (SDV)*.

## B. Construction of Conference-Level Measurements

**Purpose.** This appendix documents the data sources, inclusion criteria, and counting procedures used to construct the conference-level graphs reported in §2.3. It introduces no new claims or interpretations. Its sole purpose is to make the descriptive measurements referenced in the main text auditable and reproducible.

**Scope.** The procedures described here support the empirical observations in §2.3, which characterize how synthetic data is positioned relative to privacy-related language in recent machine learning research. All measurements are descriptive and do not evaluate the correctness, safety, or adequacy of any individual paper.

### B.1. Conference Corpus

We analyze accepted papers from three major machine learning conferences: ICML, NeurIPS, and ACL. Table 2 summarizes the total number of papers included from each venue and year.

For ICML and NeurIPS, all accepted papers were included regardless of presentation format (poster, spotlight, or oral). No distinctions were made between these categories for the purposes of measurement. For ACL, which publishes a single unified proceedings track, all accepted proceedings papers were included.

#### B.1.1. ACCEPTANCE FILTERS USED

Our measurement corpus represents a well-defined fraction of the accepted papers at each conference, determined by the availability of structured acceptance labels and publicly accessible proceedings. The following filters specify the acceptance categories included.

**NeurIPS (OpenReview).** Accepted papers were selected using OpenReview `accepted_venue_values`, including:

- NeurIPS 2024 Poster
- NeurIPS 2024 Spotlight
- NeurIPS 2024 Oral
- NeurIPS 2025 Poster
- NeurIPS 2025 Spotlight
- NeurIPS 2025 Oral

**ICML (OpenReview).** Accepted papers were selected using OpenReview `accepted_venue_values`, including:

- ICML 2024 Poster
- ICML 2024 Spotlight
- ICML 2024 Oral

- ICML 2025 Poster

- ICML 2025 Spotlight

- ICML 2025 Oral

**ACL (ACL Anthology).** ACL does not use OpenReview acceptance labels. For ACL 2024 and ACL 2025, papers were collected directly from the official ACL Anthology proceedings pages corresponding to each conference year.

No additional filtering was applied based on paper topic, research area, methodology, claimed contribution, or stated motivation.

### B.1.2. TOTAL PAPERS RETRIEVED

As the following table shows, the actual used corpus are largely aligned with size of the official proceedings, despite small differences, potentially attributed to how papers are linked to the keywords used in B.1.

*Table 2.* Comparison between official proceedings counts and corpus sizes used in our analysis. Minor differences reflect scope and indexing definitions rather than substantive inclusion differences.

| Conference | Year | Official Proceedings | Corpus Used |
|---|---|---|---|
| ACL | 2024 | 1,035 | 1,040 |
| ACL | 2025 | 1,963 | 1,977 |
| ICML | 2024 | 2,609 | 2,610 |
| ICML | 2025 | ∼3,250 | 3,257 |
| NeurIPS | 2024 | 4,034 | 4,034 |
| NeurIPS | 2025 | ∼5,290 | 5,287 |

### B.2. Synthetic-usage tiered detection

We obtain the notion of *synthetic data usage* using a tiered keyword framework that distinguishes explicit generation claims from weaker or contextual mentions. All tiers are implemented using case-insensitive regular expression matching over the full paper text.

**Tier S3: Explicit Synthetic Data.** Tier S3 captures direct and unambiguous references to fully synthetic or model-generated datasets. A paper is labeled s3_hit if it contains any of the following phrases:

> "synthetic data", "synthetically generated data", "synthetic dataset", "generated dataset", "LLM-generated data", "model-generated data", "self-generated data", "self-instruct", "instruction synthesis", "synthetic instruction(s)", "data augmentation via LLM", "LLM augmentation".

These phrases are treated as explicit claims that the dataset itself is synthetically generated.

**Tier S2: Implicit Synthetic or Programmatic Data.** Tier S2 captures weaker but still substantive indicators of synthetic data usage, such as pseudo-labeling or programmatic generation. A paper is labeled s2_hit if it contains any of the following phrases *and* the phrase occurs within a fixed token window of a data noun (e.g., "data", "dataset", "samples", "labels"):

> "generated examples", "pseudo-labels", "pseudo labels", "self-training labels", "teacher-generated", "weak supervision generated by LLM", "bootstrapped dataset", "programmatically generated dataset", "artificial data".

This contextual requirement avoids classifying generic mentions of generation that are unrelated to dataset construction.

**Tier S1: Broad Synthetic Mentions.** Tier S1 captures broad or ambiguous references to synthetic generation, including the terms:

> "synthetic", "data generation", "sample".

This tier is recorded for completeness but is not used to define synthetic data usage in aggregate statistics.

**Derived Indicator.** A paper is considered to *use synthetic data* if either s3_hit or s2_hit is true.

### B.3. Simulation Detection

To distinguish simulation-based evaluation from synthetic data generation, we separately detect simulation language.

- **simulation_hit.** A paper is labeled simulation_hit if it contains any inflection of *simulation*, *simulate*, *simulates*, *simulating*, *simulated*, or *simulator*.

- **simulation_data_context.** This flag is set when a simulation term appears within a fixed token window of both (i) a data noun (e.g., *data*, *dataset*, *samples*, *labels*) and (ii) a generative verb (e.g., *generate*, *produce*, *create*, *synthesize*).

- **simulation_only.** A paper is labeled simulation_only if simulation language is present, but no Tier S2 or Tier S3 synthetic indicators are detected. This distinction prevents conflating simulated environments with synthetic datasets.

### B.4. Privacy-Related Variables

We identify privacy claims and mechanisms using multiple keyword categories, each corresponding to a distinct form of privacy reasoning.

- **privacy_claim.** A paper is labeled privacy_claim if it contains general privacy-related language, including *privacy-preserving*, *privacy safe*, *safe to share*, *avoid sharing sensitive data*, *data sharing restrictions*, *anonym\**, *de-identif\**, *GDPR*, *HIPAA*, *personal data*, *PII*, *confidential*, *sensitive attributes*, or *no real user data*.

- **has_formal_privacy.** This indicator captures explicit references to formal privacy mechanisms, including *differential privacy*, *DP-SGD*, *Rényi DP*, *privacy budget*, *epsilon*, *delta*, *Gaussian mechanism*, *Laplace*, *PATE*, *privacy accountant*, or *moments accountant*.

- **has_privacy_audit.** A paper is labeled has_privacy_audit if it references adversarial privacy evaluation methods such as *membership inference*, *MIA*, *model inversion*, *reconstruction attack*, *attribute inference*, *singling out*, *linkability*, *exposure metric*, *canary*, or *training data extraction*.

- **has_scrubbing.** This flag captures dataset-level sanitization techniques, including *PII scrubbing*, *PII removal*, *redaction*, *de-identified via regex*, *NER-based removal*, *remove emails*, *remove phones*, *remove addresses*, *sanitization*, or *data sanitization*. If explicit scrubbing terms are absent, scrubbing is also inferred when safety-filter language (e.g., *content filter*, *safety filter*) co-occurs with a personal-data reference.

- **privacy_via_access_controls.** A paper is labeled privacy_via_access_controls if it references non-technical privacy protections such as *secure enclave*, *federated access*, *data never leaves*, *on-device*, *access control*, *trusted execution environment*, or *data use agreement*.

We derive higher-level labels from these primitives. In particular, label_privacy_by_generation identifies papers that use synthetic data and invoke privacy claims without referencing formal privacy mechanisms, privacy audits, or access controls. This label operationalizes appearance-based privacy claims grounded solely in data generation.

