# OpenReview forum: "Position: Privacy Is a Claim, Not a Property of Synthetic Data"
_ICML.cc/2026/Position_Paper_Track — ICML 2026 Position Paper Track regular_

### Official Review · Reviewer_qWKy · 2026-03-08

**Significance:** 4
**Argument Clarity:** 4
**Rating:** 4
**Confidence:** 4

**Questions:**

There are three important questions related to the suggestions for improvement. I list them below.

1. Can the authors conduct some empirical examples of privacy leakage in synthetic datasets that demonstrate the risks outlined in the paper?

2. What form might a standardized evaluation protocol for synthetic data privacy claims take in practical application?

3. Under what scenarios can synthetic data reliably provide strong privacy protection, and what specific conditions must be met for this to occur?

**Alternative Views Section:**

Yes

**Compliance With Llm Reviewing Policy A Conservative:**

Affirmed.

**Discussion Potential:**

4

**Final Justification:**

I am (partially) happy with the answer.

**Paper Summary:**

The paper claims that privacy guarantees for synthetic data are frequently overstated, as privacy is often regarded as an intrinsic property of generated datasets rather than as a claim requiring evaluation under explicit threat models. The authors state that synthetic data may still disclose sensitive information through distributional memorization, membership inference, or other inference attacks.

They show how privacy claims are presented in both research and industry contexts, highlighting the lack of standardized evaluation criteria for assessing privacy risks in synthetic datasets. It argues that privacy assessments should explicitly specify adversarial models, attack capabilities, and evaluation procedures, rather than relying on intuitive assumptions regarding data anonymization.

The authors recommend transitioning to threat-model-based privacy evaluation and suggest that the machine learning community adopt more rigorous practices when making privacy claims about synthetic data.

**Position:**

Yes

**Position In Title:**

Yes

**Related Work:**

3

**Strengths And Weaknesses:**

Two major strengths are strong positions and timely topics. The authors argue that a well-defined claim: privacy should be treated as an evaluable property rather than an intrinsic property of synthetic datasets. Synthetic data is increasingly used in privacy-sensitive domains such as healthcare and finance, making the discussion highly relevant and timely. There are other strengths, such as conceptual insight, high discussion potential, and clear writing and argumentation. In particular, the distinction between privacy guarantees and privacy claims highlights an important conceptual gap in current research and practice. The authors challenge common assumptions about the privacy of synthetic data and are likely to stimulate debate within the ML community. In addition, the argument is logically structured and accessible.

The weakness of this paper lies mainly in its experiments, implementation, and applicability. There is limited empirical evidence, and the argument is largely conceptual and would benefit from concrete empirical demonstrations of privacy leakage in synthetic datasets. While the paper calls for threat-model-based evaluation, it provides less detail on how practitioners should implement such evaluations in practice. Therefore, the possible implementation is questionable. The paper fails to distinguish between cases in which synthetic meaningful privacy protection is provided and those in which data may provide it, but does not.

There are many potential perspectives for improvement. Here, I only list three major ones: 1) To conduct empirical case studies showing instances of privacy leakage within synthetic datasets. 2) Develop a standard evaluation framework to assess privacy claims associated with synthetic data. And 3) clarify how threat-model-based privacy evaluation can be incorporated into existing ML evaluation workflows.

**Support:**

3

---

> ### Author Rebuttal · Authors · 2026-03-30
>
> We would like to thank the reviewer for their support of our work, as well as for their time and effort in reading our manuscript and providing a constructive review. We are glad the reviewer found the position clearly argued and timely. We address the weaknesses, questions, and concerns below.
>
> *On “limited empirical evidence” and request for “conduct empirical case studies”:*
>
> We appreciate the concern and acknowledge that empirical evidence is significant towards our claim. We want to clarify that the paper contains two empirical contributions that address this concern. The first is our conference-level audit (Section 2.3), which covers accepted papers from ICML, NeurIPS, and ACL across the 2024-2025 publication cycles. Our audit shows that among papers that use privacy-related language associated with synthetic data, only a small fraction pair such claims with any form of verification. Second, in Section 3, we synthesize concrete failure modes from the existing literature, showing that attacks such as membership inference can succeed despite standard audit checks confirming the safety of synthetic data usage (Akkus et al., 2025; Meeus et al., 2025; Yao et al., 2025). This directly addresses the concern that the risk may be speculative: the failure mode already occurs under current practices. Our contribution, as a position paper, is therefore not to introduce new attacks, but to surface and connect existing empirical evidence, and to explain why these risks are systematically under-recognized under current evaluation norms.
>
>
> *On “possible implementation is questionable” and the request for a “standardized evaluation protocol… in practical application”:*
>
> We agree that this should have been more explicit. The paper already specifies the components of such a protocol in sections 4 and 6, but lacks a uniform structure that allows practitioners to go through it in checklist-style.
>
> In the revision, we will organize Sections 4.1 and 6 into a concise “Minimum Privacy Claim Standard” for those who may seek to provide a privacy claim with their use of synthetic data,  consisting of steps:
>
> First, clearly state whether the synthetic data is used as (a) a risk-reduction heuristic, or (b) evidence supporting a privacy claim. Then, if (b), one should:
>
> 1. State a threat model or concrete threats they protect against, such as specifying the adversary or inference being bounded.
> 2. Scope the claim, such as clarifying what risks are addressed and what remains unaddressed.
> 3.  Provide one evidentiary mechanism — e.g., a DP guarantee with stated ε, an inference-based audit, or disaggregated risk reporting.
> This is analogous to shifts already adopted in ML: fairness (disaggregated reporting) and robustness (adversarial evaluation). Our proposal applies a similar principle to privacy claims. We are not seeking to bring new infrastructure or pipelines into the community, but want to use existing tools towards establishing a minimal disclosure norm.
>
>
> *On the concern that the paper “fails to distinguish between cases in which synthetic data provides meaningful privacy protection and those in which it does not”:*
>
> We want to respectfully emphasize that this paper does not argue *against* the use of synthetic data. As discussed in Sections 4.2 and 5, synthetic data is effective for augmentation, debugging, benchmarking, and access control. In these cases, no additional justification is required. Our argument applies only when a privacy claim is made. Our conference-audit evidence shows that synthetic data can be presented as evidence that privacy concerns are resolved, but our literature review indicates that, in those cases, privacy risks may remain unresolved. Therefore, we urge the community to reconsider current practices and call for a more well-structured way for practitioners to specify, themselves, when/how/where their use of synthetic data provides meaningful privacy protection in a falsifiable, well-established way. This distinction should be clearer and will be highlighted in the revised paper.

---

> > ### Author Rebuttal · Reviewer_qWKy · 2026-04-01
> >
> > The authors partially addressed two issues and barely touched the third. We will keep our shores.

---

### Official Review · Reviewer_sB56 · 2026-03-13

**Significance:** 4
**Argument Clarity:** 4
**Rating:** 6
**Confidence:** 3

**Questions:**

1. “Can be falsified” is different than “will be falsified.” E.g. for empirical inference evaluations, whether or not a misguided privacy claim is falsified depends on the dataset, what the adversary knows or has access to,  how clever the attack is, etc. If an attack succeeds then it’s falsified, if an attack fails then a privacy claim is not falsified but it’s also not verified. This is certainly better than making a vacuous privacy claim but ultimately I feel like it still offers false security. So I wonder if the authors could elaborate a little bit more about why falsifiability is a desirable requirement (vs always wanting a stronger requirement like explicit risk bounding)?

2. In Section 4.1 on explicit risk bounding, the paper states that differential privacy, "if carefully applied," can address the uneven risk allocation  issue. Are the authors distinguishing between pure and approximate DP, i.e. pure DP protects even worst-case data whereas approximate DP might permit a privacy failure for low-probability tail records?

**Alternative Views Section:**

Yes

**Compliance With Llm Reviewing Policy A Conservative:**

Affirmed.

**Discussion Potential:**

4

**Final Justification:**

The other reviewers raised some valid concerns about the paper, but ultimately I still think very highly of this paper and am keeping my score.

**Paper Summary:**

This paper calls out a shift in how synthetic data is perceived in privacy-sensitive machine learning settings: rather than treating privacy as an empirical claim based on the threat model and inference risk, many recent ML publications are now alluding to privacy as an implicit property of synthetic data. This "appearance-based" privacy may satisfy surface-level criteria for unidentifiability (e.g. absence of detectable secrets) but the generated output as well as any downstream tasks that use it may very well have a dependence on the training data that could be adversarially exploited. The authors call for researchers, organizations, and institutions to coordinate some lightweight changes in order to reframe privacy as a testable and verifiable property.

**Position:**

Yes

**Position In Title:**

Yes

**Related Work:**

4

**Strengths And Weaknesses:**

Strengths
* The authors provided strong evidence to support their claims about how synthetic data is used in privacy-sensitive settings; the audit summarized in Figures 1 and 2 is remarkably well-done. I also thought the call to action was really well-scoped and succeeds in being both specific and pragmatic.
* The paper has great discussion potential -- for example, while looking at the very orange NeurIPS bar, I found myself wondering whether mandatory ethics statements could be one of the inadvertent causes of privacy being over-promised.
* Overall I found the paper to be highly compelling and persuasive with a clear mission, claims all supported by concrete evidence, and the potential for good impact.

Weaknesses
* I felt like I could have used a little more help interpreting Figures 1 and 2, in that clearly blue is good and orange is bad but I couldn't decide how gray fit in with the paper's message. (Is using synthetic data with no privacy framing good, because it doesn't make false claims; or bad, because the privacy claims may be implicit?)

**Support:**

4

---

> ### Author Rebuttal · Authors · 2026-03-30
>
> We thank the reviewer for their generous assessment and encouraging feedback. We are glad to address the questions they raised to sharpen the argument for this paper.
>
> *On figure interpretation, what do the gray bars mean?*
>
> This is a fair and important point that we will clarify more in our final version. The gray category, using synthetic data without privacy framing, is neither good nor bad, and its interpretation depends on context. Papers could use synthetic data purely for augmentation, debugging, or benchmarking without any privacy-related motivations or claims. Therefore, they fall out of the discussion range of this position paper. We acknowledge that our rule-based screening can miss some papers in this category. The gray bar papers may implicitly claim privacy protection, but do not use any of the keywords in our pipeline. However, we intentionally used a stricter keyword set to avoid false positives. The gray bar here represents the ambiguous set of papers that use synthetic data, for which our current evaluation metrics cannot decisively determine whether they are “good” or “bad”.
>
> *Q1: Why falsibility as a desirable requirement rather than explicit risk bounding?*
>
> The reviewer identifies a hierarchy we want to be precise about. What we tried to claim is that falsifiability should be the minimum necessary condition, rather than a sufficient one, and explicit risk bounding is preferred when feasible. We advocate for falsifiability as the floor because, currently, many claims via synthetic data are non-falsifiable by design. The claim is made with no adversary, no success condition, and no scope. Restoring falsifiability should be the minimum requirement to make such privacy claims evaluable and comparable. A stricter bound should be preferred in many cases, but we also don’t want to make our position a verification burden for ML practitioners, since some privacy guarantees (such as DP) can be impractical at times.
>
> *Q2: Are you distinguishing between pure and approximate DP regarding tail record protection?*
>
> Yes, this is precisely the “if carefully applied” intended to signal, and we appreciate the reviewer highlighting it. Pure DP provides worst-case protection for all records, including low-density tail records, whereas approximate DP permits a δ probability of privacy failure; in practice, that failure probability can concentrate in the tail and on minority records we identified as most exposed in Section 3.2. We didn’t go into the exact details of DP application since it’s one of the existing tools we quote as useful for the question at hand, but we can make this distinction more explicit in the final version.

---

> > ### Author Rebuttal · Reviewer_sB56 · 2026-04-04
> >
> > I thank the authors for their rebuttal; I will keep my score.

---

### Official Review · Reviewer_HXbv · 2026-03-13

**Significance:** 3
**Argument Clarity:** 4
**Rating:** 4
**Confidence:** 4

**Questions:**

1. Do you see differential privacy as the preferred gold standard, or just an example of explicit risk-bounding among several acceptable approaches?

2. How do you distinguish between papers that merely use synthetic data in a privacy-sensitive area and papers that are actually making privacy-relevant claims?

**Alternative Views Section:**

Yes

**Compliance With Llm Reviewing Policy A Conservative:**

Affirmed.

**Discussion Potential:**

3

**Final Justification:**

The authors have well-addressed my concerns in their rebuttal.

**Paper Summary:**

This paper’s position is that privacy should not be treated as an inherent property of synthetic data, but rather as an explicit and testable claim about residual inference risk under specified assumptions. The authors argue that in recent ML research and deployment, privacy has gradually shifted from an adversarial, threat-model-based concept to an interpretation based on appearance. Here, synthetic data is often assumed to be privacy-preserving because it is generated rather than directly copied, lacks one-to-one correspondence with real records, or appears novel. The paper argues that this shift is a change in community standards for what privacy evidence counts as sufficient. The paper has done a conceptual analysis of papers from recent conferences (ICML, NeurIPS, and ACL). They argue that synthetic data is frequently used in privacy-sensitive contexts without explicit threat models, privacy language is often present, but substantive privacy verification is comparatively rare. This paper then discusses and develops 3 main consequences of such a shift: (1) false negatives become frequent, (2) privacy risk is redistributed rather than eliminated, and (3) privacy becomes institutionally non-actionable. In response, the paper suggests re-grounding privacy as an explicit empirical claim, synthetic data should not automatically inherit privacy status and instead be earned through scoped and testable evidence.

**Position:**

Yes

**Position In Title:**

Yes

**Related Work:**

3

**Strengths And Weaknesses:**

# Strengths

1. The paper is conceptually clear. It has elaborated on the differences between privacy as adversarial inference risk and privacy as an appearance- or generation-based property. The discussion is also clear regarding how output-centric reasoning can be intuitively appealing while still being structurally insufficient.

2. Despite being only descriptive, the empirical audit is also valuable. It attempts to document a pattern in research practice rather than relying purely on anecdotal impressions. The Figures in section 2.3 also supports the paper’s claim regarding privacy language often appears without corresponding explicit verification.

3. The discussion is also strong regarding the failure modes. The sections on false negatives, tail-risk concentration, and institutional non-actionability are effective because they connect technical issues with downstream governance and accountability consequences.

# Weaknesses

1. The empirical analysis appears a bit limited by its rule-based screening design. The argument depends heavily on how privacy claims are being framed, so interpretation matters a lot. It would help to know more about false positives and false negatives in the coding process, how indirect claims were distinguished from explicit ones, and whether some papers were manually reviewed for calibration. In the current manuscript, the audit is promising but not yet fully convincing as strong empirical backing.

2. The discussion of institutional and legal consequences is insightful, but a bit broad. The paper moves between ML publication norms, vendor claims, etc. While these are clearly connected, each arena operates differently. A bit more differentiation would strengthen the analysis.

3. The paper occasionally risks understating the practical value of weaker evidence in applied settings. In many real deployments, formal privacy guarantees may be infeasible, threat models may be underspecified, and negative empirical evidence may still represent meaningful progress relative to raw data sharing. The paper acknowledges this in the alternative views section, but the main narrative sometimes leans toward a standard sharper than some applied domains can realistically meet.

**Support:**

3

---

> ### Author Rebuttal · Authors · 2026-03-30
>
> We thank the reviewer for the careful and constructive review. We are glad the reviewer found the paper conceptually clear and the empirical audit valuable, and we will address the weaknesses and questions directly below.
>
> *On the "rule-based screening design" and calibration,*
>
> We appreciate this concern and want to clarify that calibration considerations were central to the design, even though they were not reported as a separate analysis. First, the pipeline is deliberately conservative in scope due to its rule-based nature. For example, the *privacy_by_generation* label, which indicates whether a paper makes an appearance-based privacy claim, must simultaneously satisfy the synthetic data usage criterion, the presence of a privacy claim, and the absence of formal mechanisms, audits, or access controls. Specific rules and keywords are in Appendix B. We acknowledge that rule-based screening of the language in papers is prone to errors; however, we designed the system to be conservative, thereby more likely to undercount appearance-based claims. Additionally, the purpose of this empirical study is to highlight that the problem we are addressing is not imaginary and is increasingly prevalent in the ML research community, rather than seeking perfect statistics from previously published academic papers.
>
> *On the discussion being "a bit broad" across institutional arenas:*
>
> We agree with the point and want to clarify that the breadth was intentional. Section 2.2 moves across industry, legal, and research contexts, not treating them as equal risks, but to show the mutual reinforcement across these fields. We acknowledge that different arenas may hold different practices towards privacy claims, but the main narrative here is to show how they sustain and enhance the norm of appearance-based privacy claims. We agree that a sharper distinction may be beneficial, but the breath here is analytically motivated.
>
> *On understating "the practical value of weaker evidence in applied settings",*
>
>
> We thank the reviewer for pointing out that this is addressed in Alternative View 2 (Section 5), which acknowledges that negative empirical results “may be the most tractable signal available” in applied settings and represent meaningful progress in many cases. Our response addresses why our concern is about the institutional role this evidence has come to play, especially when the absence of observed leakage is increasingly treated as a resolution rather than a provisional signal. What we attempted to show with our empirical evidence is that the use of synthetic data often serves as an implicit claim that privacy concerns are resolved, without specifying the scope, the adversary, or the residual risks. The problem is not the strength of the evidence, but unscoped privacy claims. What we hope for is not that every synthetic data application adopts a strict privacy audit, but that when a privacy claim is made, readers can understand what is protected rather than infer it from the use of synthetic data. We hope this clarifies that our position is about responsible claims, not the burden of verification.
>
>
> On Q1: *Is differential privacy the preferred gold standard, or one example among several acceptable approaches?*
>
> DP is an example, not a golden standard. We understand that in many application scenarios it’s neither practical nor feasible to achieve a DP guarantee (as we state in the paper, “DP is not universally applicable”), and we present DP here as an example of how privacy claims can be articulated in terms of residual risks. In the examples we provide, DP is indeed one of the better-structured, theory-backed solutions, if that is preferred by the researchers in need.
>
>
> On Q2: *How do you distinguish papers merely using synthetic data in a privacy-sensitive area from papers actually making privacy-relevant claims?*
>
> This distinction is specified in Appendix B. The privacy claim language consists of “privacy preserving”, “GDPR-compliant”, etc. Papers using synthetic data in a privacy-sensitive domain without invoking any privacy language fall into the "no privacy framing" category shown in Figure 2 and are explicitly excluded from the appearance-based claim count. We acknowledge that rule-based language screening can make mistakes, and we understand the concern about false positives for papers that use synthetic data and make privacy claims. However, it is infeasible to manually check all papers in the empirical audit, and the current rules are intentionally calibrated to be strict, so we are more likely to undercount rather than overcount.

---

> > ### Author Rebuttal · Reviewer_HXbv · 2026-04-02
> >
> > The authors have well-addressed my concerns in their rebuttal.

---

### Official Review · Reviewer_tpHd · 2026-03-21

**Significance:** 4
**Argument Clarity:** 3
**Rating:** 5
**Confidence:** 4

**Questions:**

It would be good if the authors could go in more depth in the approaches or frameworks that they would recommend to follow for the privacy auditing of synthetic data.

**Alternative Views Section:**

Yes

**Compliance With Llm Reviewing Policy A Conservative:**

Affirmed.

**Discussion Potential:**

4

**Paper Summary:**

This position paper aims at showing that the use of synthetic data is currently done in many ML contexts without clearly defining the adversary models and formalizing the privacy guarantees achieved. In addition, the paper also advocates that current approaches for quantifying the privacy of synthetic data often fails to capture meaningful privacy risks. Some approaches for remediating to this situation are also discussed.

**Position:**

Yes

**Position In Title:**

Yes

**Related Work:**

4

**Strengths And Weaknesses:**

The paper is well-written but it would have helped to specify the main contributions of the paper in the introduction. The authors clearly discussed why the use of synthetic data cannot be considered as preserving privacy on its own and why simple approaches such as measuring the number of replicas cannot be considered as a useful privacy metric. The authors have also clearly reviewed how companies providing synthetic data as a core product advertise the privacy of their solutions, without formal explicit audit or evaluation.

The studies of recent conference papers also demonstrate that synthetic data has become an integral part of many machine learning pipelines but that very few papers effectively conduct a rigorous privacy evaluation despite the privacy discourse existing in the paper. The resulting negative consequences of this discourse, such as the fact that other privacy attacks remain possible or that rare or atypical records remain vulnerable. The paper also discussed possible directions to improve the privacy evaluation such as explicitly bounding the inference risks through the use of differential privacy, the use of benchmarking framework for assessing these privacy risks.

One of the limitations of the paper is that it stays at a relatively high level in terms of approaches proposed for the audit of synthetic data. It would have been good to go a bit in more depth in the solutions that could be adopted in the call to actions.

**Support:**

3

---

> ### Author Rebuttal · Authors · 2026-03-30
>
> We thank the reviewer for the positive assessment and for highlighting the timely significance of our position and the strengths of our empirical findings.
>
> On the suggestion to *“specify the main contributions of the paper in the introduction,”* we agree and will revise the introduction to more explicitly summarize the paper’s core contributions and positioning.
>
> Regarding the request to *“go in more depth in the solutions that could be adopted,”* we appreciate this suggestion and agree that concrete solutions are vital to addressing our position. While our goal is not to introduce a new auditing framework, we agree that the practical implications can be made more concrete. In the revision, we will reorganize the existing solutions proposed in Sections 4 and 6 to provide a clearer structure that clarifies how existing tools (e.g., threat model specification, inference-based evaluation, and risk disaggregation) can be applied in practice. Our goal is to surface it as a concrete, named standard rather than a discursive recommendation, so that authors and reviewers can find the most appropriate tools to address their concerns.

---

> > ### Author Rebuttal · Reviewer_tpHd · 2026-04-04
> >
> > I thank the authors for the rebuttal. As mentioned in my review, if the paper is accepted I suggest to go in more depth in the description of the solutions that could be adopted.

---

### Decision · Program_Chairs · 2026-04-30

**Decision:**

Accept (regular)

**Comment:**

My decision is to accept the paper.

The paper argues that privacy claims wrt synthetic data have become "watered down" in recent ML research and practice, such that the appearance of privacy underlies most claims, rather than formalizing inference risks under certain threat models. The paper provides empirical support for this claim about trends, and considers the implications of allowing this shift to stick. The paper recommends shifts in norms wrt privacy language at several levels.

Reviewers agreed that the paper makes a compelling argument about shifting norms and the issues that can arise as a result. There were some concerns about concreteness in the call to action, which were addressed in rebuttals. Adding the Minimal Privacy Claim Standard specification to the paper would address most of the objections.